# Bayesian Gaussian Mixture Models for Enhanced Radar Sensor Modeling: A Data-Driven Approach towards Sensor Simulation for ADAS/AD Development

**DOI:** 10.3390/s24072177

**Published:** 2024-03-28

**Authors:** Kelvin Walenta, Simon Genser, Selim Solmaz

**Affiliations:** 1Virtual Vehicle Research GmbH, Inffeldgasse 21a, 8010 Graz, Austria; kelvin.walenta@v2c2.at (K.W.); selim.solmaz@v2c2.at (S.S.); 2Institute of Theoretical and Computational Physics, Petersgasse 16, 8010 Graz, Austria

**Keywords:** virtual validation, sensor simulation, ADAS/AD development, Gaussian mixture models, radar modeling, data-driven modeling

## Abstract

In the realm of road safety and the evolution toward automated driving, Advanced Driver Assistance and Automated Driving (ADAS/AD) systems play a pivotal role. As the complexity of these systems grows, comprehensive testing becomes imperative, with virtual test environments becoming crucial, especially for handling diverse and challenging scenarios. Radar sensors are integral to ADAS/AD units and are known for their robust performance even in adverse conditions. However, accurately modeling the radar’s perception, particularly the radar cross-section (RCS), proves challenging. This paper adopts a data-driven approach, using Gaussian mixture models (GMMs) to model the radar’s perception for various vehicles and aspect angles. A Bayesian variational approach automatically infers model complexity. The model is expanded into a comprehensive radar sensor model based on object lists, incorporating occlusion effects and RCS-based detectability decisions. The model’s effectiveness is demonstrated through accurate reproduction of the RCS behavior and scatter point distribution. The full capabilities of the sensor model are demonstrated in different scenarios. The flexible and modular framework has proven apt for modeling specific aspects and allows for an easy model extension. Simultaneously, alongside model extension, more extensive validation is proposed to refine accuracy and broaden the model’s applicability.

## 1. Introduction

As the landscape of ADAS/AD (Advanced Driver Assistance and Automated Driving) systems evolves, the role of virtual testing becomes increasingly significant. Virtual testing, a methodological breakthrough, allows for the extensive and meticulous evaluation of AD systems in simulated, yet highly realistic environments. This process enables the emulation of diverse and challenging real-world scenarios, which are often impractical or hazardous to replicate in physical testing. By leveraging advanced simulation technologies, virtual validation provides a safer, more efficient, and cost-effective means of assessing the performance, reliability, and safety of ADAS/AD functions.

The testing and homologation of ADAS/AD functions demand extensive testing efforts, involving millions of kilometers driven for verification and validation purposes, as demonstrated in references such as [1,2]. Realistically achieving this level of testing through physical means is deemed impossible and is consequently not pursued by any major OEM. Various alternative strategies are in place to streamline the process, which are all centered around simulation, ultimately leading to virtual validation and homologation. However, it is important to note that this does not imply that everything is conducted solely in virtual environments. In fact, many approaches integrate both virtual and physical tests, and sometimes even hybrid testing methods [3,4].

For performing simulations for the virtual validation of ADAS/AD functions, the coupling or integration of different simulation tools is typically required, as not all relevant simulation models are available in one simulation tool. Therefore, a generic and basic simulation framework typically consists of four major submodels: (1) the Vehicle model, (2) the Environment simulation, (3) the Sensor model, (4) the ADAS/AD function, as depicted in Figure 1.

The sensor models represent the subsystem responsible for perception. Typically, distinct sensor models are employed for various perception hardware, such as cameras, radar, and lidar.

### 1.1. State of the Art in Perception Sensor Modeling

Perception sensor models can be classified according to different aspects. For example in [6], the proposed categories are called low-, medium-, and high-fidelity sensor models, implying that the accuracy of the models depends on their modeling technique. This assumption can be misleading, and therefore the authors of this paper prefer a distinction into geometrical, statistical, and physical sensor models. It might often be the case that a physical sensor model is more accurate than a statistical one, but as long as it is not proven, the naming should not lead to a possible wrong impression.

Additionally, sensor models are distinguished by the output they are generating, either so-called object-list sensor models or raw data sensor models. In object-list sensor models the sensor itself and the perception, object detection, and object tracking software are modeled, and therefore those models generate a list of tracked objects, such as vehicles, pedestrians, houses, and so on. Such sensor models are often used for the testing and development of the planning and control aspects of ADAS/AD functions. The object lists are typically of very similar structure for camera [7,8,9], lidar, and radar [10], usually containing basic information about the size, position, and speed of the objects. A geometric model for example, focusing on occlusion effects, can be found in [11] and a general approach for object-list models is stated in [12].

For so-called raw data sensor models, the differences in output for different sensors are much more pronounced. For example, a lidar model [13,14] typically generates point clouds while a camera model produces images. The main difference is that in a raw data sensor model, only the sensor itself is modeled and not its software, with interfaces very similar to the real sensor. However, this distinction is not always clear-cut, as raw data often undergo significant post-processing, sometimes yielding object lists as output. Raw data sensor models are typically used for generating synthetic data for the testing and development of perception, tracking, or detection algorithms, rather than being utilized extensively in closed-loop simulations for validating complete AD functions. Many environment simulation software such as aiSim [15], Dyna4 [16], Carmaker [17], VTD [18], and Carla [19] offer built-in sensor models for different sensor types and outputs. Nonetheless, the field of external sensor models is growing as well, as individual parameterization is often preferred. To explore standardization efforts concerning sensor model databases aligned with the open simulation interface (OSI) within the ADAS context, refer to [20].

### 1.2. Scope of This Work

Radar sensors play a pivotal role in modern ADAS and AD applications, providing crucial information about the surrounding environment. One fundamental aspect of radar sensors is the accurate modeling of radar cross-section (RCS), which directly influences sensor performance and perception accuracy. In this study, we propose a novel modeling approach for object-list-based radar sensor models, emphasizing the detailed representation of RCS and its integration with an existing geometric sensor model. This is achieved by employing a modular framework composed of distinct components; namely, an angular Field-of-View filter and Gaussian mixture models (GMMs), to model the sensor’s perception of vehicles. By focusing on RCS modeling within the context of object-list-based radar sensor simulations, our novel approach aims to improve the fidelity and realism of radar sensor simulations in ADAS and AD environments.

### 1.3. Structure of the Article

In the next section, the radar sensor model will be introduced and described in detail. Section 3 consists of a description of the required measurement data and training of the sensor model. In Section 4, the results for the RCS modeling and the object-list output in general are stated and discussed. The paper concludes with a summary, conclusion, and future work in Section 6.

## 2. Radar Sensor Model

The radar sensor model outlined in this study relies on object lists and comprises various components illustrated in Figure 2. The model takes as input an object list, encompassing comprehensive information for each object, including dimensions, positions, velocity, orientation, and more. Subsequently, the succeeding sections will elaborate on each step of the model, with particular emphasis on the RCS model, as it constitutes the central aspect of this research.

### 2.1. Initial Field-of-View Filter

Initially, the object list undergoes a filtering process based on position and orientation. The aim is to minimize the number of objects taken into account for radar sensor modeling. This filtering is determined by the radar sensor’s initial Field-of-View, which is pre-defined prior to the simulation. The Field-of-View encompasses an angular range in azimuth and elevation, along with a maximum threshold for the range. Any objects positioned outside this specified Field-of-View are excluded. The objects that pass this screening serve as input for the subsequent radar cross-section model.

### 2.2. Radar Cross-Section Modeling

The modeling of a radar sensor, and more specifically, the radar cross-section (RCS) requires mimicking the perception of the radar as realistically as possible. As many radar sensors in the automotive industry perceive targets as a distribution of scatter points, each with an individual RCS value, the model needs to be able to reconstruct such a cluster point distribution.

#### 2.2.1. Cluster Point Modeling

The foundation of the proposed model is based on Gaussian mixture models (GMMs). A GMM is a probabilistic model that assumes that all the cluster points originate from a mixture of a finite number of Gaussian distributions of the form
(1)$$p\left(x\right)=\sum_{k=1}^{K}\pi_{k}N\left(x|\mu_{k},\Sigma_{k}\right).$$

The parameters $\pi_{k}$, $\mu_{k}$, and $\Sigma_{k}$ are inferred from the data [21]. To incorporate external influences on the cluster point distribution, such as the aspect angle, the following framework is suggested: The angular range covered by the model is discretized by a fixed set of supporting points. Each supporting point contains a Gaussian mixture model as well as a respective aspect angle. This is schematically depicted in Figure 3.

This approach enables the framework to function as a generative model for any types of data that consist of cluster points. By providing an aspect angle as input, the model aims to replicate the cluster point representation of the trained object for that specific aspect angle. To achieve this, it creates samples from the closest supporting points to the input angle. The number of samples per frame is determined by sampling from the histogram depicting the frequency of detections per frame in the training data. A minimum distance between the individual cluster points is enforced by re-sampling a point in case it is too close to the rest of the points.

#### 2.2.2. Parameter Inference—A Bayesian Approach

For the inference of the parameter in the GMMs, different methods exist. One common approach is to determine the set of parameters that maximize the likelihood function under the constraint that all weights sum up to one. This results in a set of equations that have to be solved iteratively, an approach that is referred to as the Expectation-Maximization algorithm [21]. One major disadvantage of this approach is the fact that the number of components *K* in Equation (Equation 1) has to be chosen a priori. Choosing it too low might result in an insufficient fit to the data. In contrast, taking a higher number of components than necessary will provide a better fit during training, but generalization to unseen data is usually worse. Therefore, it is desirable to infer the number of components from the data without specifying the exact value in advance.

For this purpose, a Bayesian approach poses an elegant solution. In the following, the main idea behind this approach will be described without going into detail. For the full derivation, including proofs and examples, the reader is referred to the work of Bishop et al. [21] as well as Blei et al. [22]. Instead of calculating the maximum-likelihood estimate of the parameter, one calculates the posterior distribution of all latent variables, denoted as Z, given the data X:(2)$$p\left(Z|X\right)=\frac{p(X|Z)p(Z)}{p(X)}.$$

The calculation of the marginal p(X) is usually intractable, and approximation schemes become necessary. These are usually divided into two categories: stochastic approximations, as for example Markov chain Monte Carlo, and deterministic approaches, as the variational inference algorithm used in this work. The main idea is that one can decompose the log of the marginal in the following form [21]:(3)lnp(X)=L(q)+KL(q||p).
with
(4)$$L\left(q\right)=\int q\left(Z\right)\ln\left\{\frac{p(X,Z)}{q(Z)}\right\}dZ$$
(5)$$\mathrm{KL}\left(q||p\right)=-\int q\left(Z\right)\ln\left\{\frac{p(Z|X)}{q(Z)}\right\}dZ$$

The first term represents the Evidence Lower Bound (ELBO) of the variational distribution *q* and the second term is the Kullback–Leibler divergence between *q* and the true posterior. Thus, instead of calculating the posterior distribution, one can maximize the ELBO with respect to the variational distribution *q*. Without any restrictions, maximizing the ELBO would result in the true posterior distribution [21]. In order to make the calculations tractable, a factorization of q(Z) between the latent variables is enforced, which is an approach that is known as mean-field approximation [22]. An implementation of the algorithm is included in the *scikit-learn’s* library [23] in Python and used in this work. The parameters that need to be set for each mixture model are the prior distributions and number of components. For the complete model, the amount of training data for each supporting point as well as the number of supporting points need to be chosen.

### 2.3. Occlusion and RCS-Based Detectability

As a first step towards modeling occlusion, one needs to determine how the objects inside the Field-of-View (FOV) occlude each other. Therefore, an occlusion matrix is generated. The occlusion matrix holds Boolean values and gives two outputs for each object. The row defines which objects are occluded by the current object, and the column defines which objects are occluding the current object. For more details, refer to Genser et al.’s original work on the occlusion model [11]. To illustrate, consider the example scenario below, which shows two vehicles in the Field-of-View of the ego vehicle. The second vehicle cuts the lane of the first vehicle and is therefore partially occluded. Using the RCS model, the process begins by creating a scatter point representation of the occluded vehicle. Subsequently, all points that are occluded are eliminated (see Figure 4). The remaining contributions are then summed up to determine the total RCS.

Given that the radar cross-section determines an object’s capability to reflect radar signals back to the receiver, its detectability is inherently tied to the RCS. The radar equation provides a straightforward approximation to link detectability with the RCS:(6)$$P_{r}=\frac{P_{t}G_{t}G_{r}\lambda^{2}\sigma }{{\left(4\pi \right)}^{3}R^{4}}$$
with the received power $P_{r}$, the transmitted power $P_{t}$, the gains $G_{t}$ and $G_{r}$ of the transmitting and receiving antennas, the wavelength λ, the range *R* to the target, and the radar cross-section σ [24]. Assuming that the RCS only affects the maximum range for the detection of an object, one can use this equation to find a simple model for the FOV given an arbitrary RCS value. By rearranging the terms and introducing the minimum detectable power by the receiver $S_{min}$ as the received signal power, it becomes possible to solve for the maximum range $R_{m}$ corresponding to an arbitrary RCS value σ:(7)$$R_{m}=\sqrt[{4}]{\frac{P_{t}G_{t}G_{r}\lambda^{2}\sigma }{{\left(4\pi \right)}^{3}S_{min}}}=K\cdot \sqrt[{4}]{\sigma }$$

Here, all the radar parameters were combined into the constant *K*. Knowing the maximum range $R_{m,ref}$ for a given RCS value $\sigma_{ref}$, one can calculate the FOV for an arbitrary RCS value:(8)$$K=\frac{R_{m,ref}}{\sqrt[{4}]{\sigma_{ref}}}=\frac{R_{m}}{\sqrt[{4}]{\sigma }}\;\Rightarrow \;R_{m}=R_{m,ref}\cdot \sqrt[{4}]{\frac{\sigma }{\sigma_{ref}}}$$

### 2.4. KDE+ Correction

In the final step, the model allows for the coordinates of the objects in the model output to be corrected for sensor-specific artefacts such as noise and offsets. This is achieved by combining a stochastic component, the so-called Kernel Density Estimation (KDE), with a deterministic component (+), a regression model. While the KDE part contains the information about noise, the deterministic component is used to model offsets. The interested reader is referred to the original work by Genser et al. [7] for a detailed description.

## 3. Dataset and Model Training

In the following, crucial effects essential for realistic radar sensor modeling are investigated utilizing an open-source dataset. The subsequent step involves detailing how the model, as outlined in Section 2.2, is trained using this dataset.

### 3.1. Observed Effects

The dataset used in this work is an open-source dataset, provided by the Institute of Automotive Engineering of TU Darmstadt. It contains measurements conducted on the August Euler Airfield in Darmstadt, Germany, where a sinusoidal slalom course with ten periods was set up. The object of interest (OOI) drives through the slalom course while the ego vehicle follows it in a straight line. A schematic representation of the setup is depicted in Figure 5.

In this way, aspect angles similar to many real-world traffic scenarios are obtained. The experiment was conducted with different vehicles and repeated several times each. In the case of this work, the data for a BMW 535 were used, whereby one run is used for training and another run for validation. The reader is referred to the associated publication from Elster et al. for more details regarding the experimental setup and data [25]. The final dataset consists of a set of measurements, denoted as frames, that contain all the detections of the object. These detections are characterized by a position in 2-D space (*x* and *y*) as well as a corresponding radar cross-section value σ. For the purpose of model training, the coordinates are transformed into the OOI’s reference system ($x_{loc}$ and $y_{loc}$); see Figure 6 below.

The linear sum of the RCS values of all the detections on the object is considered to be the total radar cross-section Q(σ):(9)$$Q\left(\sigma \right)=\sum_{i=1}^{N}\sigma_{i}.$$

To investigate the angular dependency of the radar cross-section, the total RCS is calculated for each frame and plotted over the respective aspect angle of the frame. Due to the nature of the RCS, decibel units are used. An aspect angle of 0° corresponds to the surface normal of the OOI’s rear. This results in the RCS curve depicted in Figure 7, where a moving-median filter with a window size of 40 was applied to better illustrate the angular behavior. This corresponds to an angular interval of about 2°. As expected, the RCS profile seems to be symmetric with respect to the longitudinal rear axis. Considering the fact that the data are displayed on a logarithmic scale, there is a huge decrease in RCS as the angle relative to the longitudinal axis increases. This is, in fact, a known phenomenon, and in accordance with other literature, such as [26,27].

Besides the actual angular dependency of the total RCS, it is desirable to model the distribution of the scatter points in space, as well as their RCS value. The distribution in $x_{loc}$ is shown in Figure 8 and exhibits several peaks. These peaks are likely a consequence of the geometric and material properties of the vehicle along the x-dimension, exhibiting several different materials and shapes (e.g., rear bumper, rear window, etc.). The change in these properties in the y-dimension is far less pronounced, which results in a simpler, unimodal distribution for the $y_{loc}$ values, as is visible in Figure 9.

The distribution of the individual RCS values in Figure 10 shows a very broad spectrum, considering the logarithmic units. Since the values for the RCS varies over magnitudes, a large portion of these scatter points do not contribute to the total RCS Q(σ) significantly. However, realistic sensor modeling requires recreating model outputs that exhibit similar characteristics to the ones mentioned above.

### 3.2. Model Training

The proposed framework necessitates the definition of several key parameters as depicted in Figure 11: the prior type, the number of Gaussian components, the volume of training data for each mixture model, and the number of supporting points essential to capture the angular characteristics of the radar cross-section (RCS). The conclusive training phase is executed in Python, using the *scikit-learn* [23] library.

#### 3.2.1. Prior Selection

As the model training is performed from a fully Bayesian perspective, the prior distributions on the model parameter need to be specified. The priors over the means μ and precisions Λ are chosen to be Gaussian–Wishart, with $\beta_{0}$ and $\nu_{0}$ representing the hyperparameters of the distributions [21]:(10)$$\begin{array}{cc} p(\mu ,\Lambda ) & =p(\mu |\Lambda )p(\Lambda )\\ \; & =\prod_{k=1}^{K}N\left(\mu_{k}|m_{0},{\left(\beta_{0}\Lambda \right)}^{-1}\right)W\left(\Lambda_{k}|W_{0},\nu_{0}\right) \end{array}$$

For the prior on the weights, a Dirichlet process prior in the stick-breaking representation is used. Since the Dirichlet process can be interpreted as an infinite dimensional Dirichlet distribution, it allows for an infinite number of components. However, the algorithm infers the number of actual components active from the data. This way, only an upper bound for the components needs to be specified. For the general derivation of the variational inference algorithm, the reader is referred to the book from Bishop [21]. The derivation for the Dirichlet process prior is in the paper from Blei et al. [22].

#### 3.2.2. Number of Frames per Supporting Point

To ensure reliable parameter inference, an adequate volume of data points is crucial during the training phase. As illustrated in Figure 12, achieving this involves grouping multiple frames together. The selection of the number of frames plays an essential role in model quality. Opting for too few frames can result in a poor fit, while selecting too many may blur essential angular characteristics. Therefore, the objective is to determine the minimum data volume necessary for accurate model training.

To measure the model’s quality, a validation dataset is utilized, and the 1-D Wasserstein distance between the model output and the validation data is computed. The Wasserstein metric allows for the quantification of the similarity between two distributions. For an extensive overview of the topic, the reader is referred to the work of Panaretos et al. [28]. This process is conducted separately for each coordinate, namely $x_{loc}$, $y_{loc}$, and RCS, and for different numbers of frames. The number of frames is indirectly given by the angle interval $\Delta_{angle}$ covered within the grouped data.

The results, depicted in Figure 13, initially exhibit a significant decrease with an increased group size. However, beyond an angle interval of approximately 1.5°, the Wasserstein distance decreases only slightly. Although larger angle intervals might yield an even smaller Wasserstein distance, a value of 2° is chosen to prevent the blurring of angular characteristics while maintaining computational efficiency.

#### 3.2.3. Number of Gaussian Components

To showcase the Bayesian approach’s effectiveness, models with varying numbers of components are trained and the log-likelihoods during both training and validation are compared. This process is also replicated for the frequentist approach. The data are then plotted over the number of components and displayed in the Figure 14 and Figure 15. Using the frequentist approach, one can see that increasing the number of components results in an ever improving fit, indicated by the increasing log-likelihood. This is the expected behavior as the frequentist approach aims to maximize the log-likelihood; a more complex model enables a better fit. However, during validation, one notices a point where additional components result in a decreased log-likelihood, indicating the beginning of the overfitting regime. This appears to be the case at around five components. The less data that are used during training, the more pronounced the overfitting is.

A different behavior is observed for the Bayesian approach. As one can see, the training as well as the validation log-likelihood increase steadily until a certain number of components is reached. After that, the log-likelihood stagnates and does not increase anymore. Even though the additional components would improve the fit, the penalty for the model complexity due to the prior distributions overrules the fit improvement. For this reason, the weights for these components are set close to zero, and the effective number of components stays the same, avoiding the overfitting problem encountered with the Expectation-Maximization algorithm.

#### 3.2.4. Number of Supporting Points

The last hyperparameter of the model is the number of supporting points used. As already pointed out in Section 3.1, the radar cross-section (RCS) is highly dependent on the aspect angle, and the introduction of several supporting points aims to recreate this angular dependency. Therefore, the similarity of the RCS curves between model output and validation data will be used as a quality criterion for models utilizing different numbers of supporting points. More specifically, models with a maximum of 10 components and an angle interval of 2° will be trained with a number of supporting points ranging from 2 to 20. The area between the RCS curves is then calculated and plotted over the number of supporting points. As there are strong fluctuations, again a moving median filter is used. Additionally, different window sizes of filters are explored to assess their qualitative impact on the outcome. The analysis reveals that increasing the number of supporting points initially enhances the model’s capacity to reproduce the angular characteristics of the RCS. However, this improvement plateaus after approximately eight supporting points, as it can be seen in Figure 16.

## 4. Results

In order to verify the effectiveness of this approach, it is first investigated whether the model output shares similar characteristics to those encountered in Section 3.1. For this purpose, a single run not included in the training data will be used for validation. A model using 10 supporting points with 10 components each and an angular interval of $\Delta =2^{\circ }$ is trained. The prior distributions for the means and precisions are chosen to be Gaussian–Wishart and Dirichlet distributions. After that, simple toy scenarios are used to showcase the capability of the full-sensor model.

### 4.1. Model Validation

Initially, the total RCS over the aspect angle is examined. A total of five simulation runs are compared to the validation data to account for stochastic fluctuations.

Examining the results in Figure 17, it is evident that the RCS characteristics of the validation data and the model-generated data are in good agreement, despite deviations among the different runs. These deviations stem from the stochastic nature of the model and are actually desirable, considering the significant fluctuations observed in Figure 7.

In a subsequent step, the distribution of scatter points in space alongside their respective RCS values is investigated. Utilizing the same validation set, histograms are constructed to visualize the frequency of $x_{loc}$, $y_{loc}$, and RCS values, aiming to assess the Gaussian mixture models’ capability to capture the underlying distributions. In order to quantify the similarity between validation data and model output, the 1-D Wasserstein distance is used. The Wasserstein distance is also computed between the training data and validation data, acting as a benchmark for the inherent fluctuations observed during individual experimental runs. Both of these values are illustrated in the accompanying figure.

The results for $x_{loc}$ and $y_{loc}$ coordinates are displayed in Figure 18 and Figure 19. It is evident that the mixture models effectively represent the underlying distributions of the individual components. However, the model appears to smooth out the distribution of $x_{loc}$ values, though these differences hold no significant bearing for sensor modeling purposes. Conversely, the model output for $y_{loc}$ values closely aligns with the validation data, likely due to the inherent simplicity of the distribution, which remains unimodal in this case. In contrast, the multimodal nature of the $x_{loc}$ value distribution poses a challenge for the model to accurately learn its underlying structure. Nevertheless, in both cases, the mean values between the model output and validation data coincide almost perfectly. The disparity between the means for the $x_{loc}$ values is 0.06 m, while for the $y_{loc}$ values, it is 0.01 m. The Wasserstein distance values for the model output closely align with the reference value in terms of magnitude, indicating that the deviations are not notably larger when compared to naturally occurring fluctuations.

Likewise, the distribution of RCS values in Figure 20 closely aligns with the validation data distribution. Minor disparities arise primarily around the peak, where the model output displays a slightly smoother distribution. This smoothing effect can be attributed to the Gaussian distributions utilized in the model, generating a smoothed estimation of the actual density. However, the differences are negligible for sensor modeling purposes. In addition, the mean values almost perfectly coincide, with a difference of $0.56\;{\mathrm{dBm}}^{2}$. The values of the Wasserstein distance again suggest that the deviation is of the same order of magnitude as the naturally occurring fluctuations.

### 4.2. Test Scenarios

In the first step, the RCS-dependent Field-of-View (FOV) is illustrated. As described in Section 2.3, the radar equation serves as the foundation for establishing a correlation between the maximum detectable range and the RCS of the object, utilizing predefined reference values. A basic scenario featuring a single vehicle in the FOV is constructed. Employing a hypothetical maximum detectable range of 8 m for an RCS value of 10 ${\mathrm{dBm}}^{2}$, the vehicle is positioned at a distance of 10 m—once with a relative angle of 0° and in another instance with a relative angle of 25°.

The resulting FOV is depicted, along with a visualization of the scenario, in Figure 21. Due to the angular characteristics of the RCS, the vehicle at a relative angle of 0° demonstrates a significantly larger FOV compared to the one at a relative angle of 25°. In this specific instance, the resulting FOV for the angled vehicle is insufficient for detecting the presence of the vehicle.

In the subsequent phase, a traffic scenario featuring two vehicles within the Field-of-View (FOV) is examined. Specifically, one vehicle executes a cut-in maneuver in front of the other, leading to partial occlusion. It is expected that as the vehicle becomes sufficiently occluded, detection will become compromised. A closer inspection of the outcomes in Figure 22, showing four consecutive frames, confirms this expectation. Once the vehicle is sufficiently occluded, it is considered undetected, as indicated with a red outline.

## 5. Remarks

As a result of its data-driven foundation, the sensor model’s validity is restricted to the angles and ranges encompassed within the dataset. Furthermore, radar data typically undergo extensive post-processing and may vary across manufacturers. Therefore, it cannot be guaranteed that this model is applicable to sensor systems from other manufacturers. Nevertheless, related research indicates that generalization is possible to some extent. As of now, validation was only performed with respect to the ability to recreate the characteristics that are found within the experimental data. Comprehensive evaluation of the entire sensor model’s performance has not been conducted due to insufficient data availability. Consequently, making quantitative assertions regarding its performance relative to traditional approaches is unfeasible. However, using real-world data offers the advantage of increased realism in contrast to idealized simulations or measurements conducted in anechoic chambers. Employing the radar equation serves as a simplified approximation for the relationship between radar cross-section (RCS) and the maximum detectable range. Nevertheless, thanks to the modular framework of the sensor model, substituting the radar equation with more sophisticated models is readily achievable.

## 6. Summary, Conclusions, and Future Work

This paper introduces a comprehensive radar sensor model based on object lists, employing a modular framework composed of distinct components, beginning with an initial angular Field-of-View (FOV) filter. Subsequently, the radar’s perception of vehicles is modeled using Gaussian mixture models (GMMs). To account for angular dependencies, supporting points are introduced, each featuring a Gaussian mixture model trained on a specific angular range. A Bayesian variational approach automatically assesses model complexity based on the data. Following this, an occlusion model is implemented to eliminate shaded detections. The RCS contributions of the remaining detections are aggregated, and, in conjunction with the radar equation, determine the radar’s ability to detect the object. A final refinement involves adjusting the model output for realistic sensor artifacts, such as noise and offsets, utilizing the KDE+ approach. The model is trained on an open-source radar dataset, utilizing the *scikit-learn* library in *Python* 3.11. Employing the Bayesian approach proved successful in preventing overfitting. The performance of the proposed framework is evaluated through a traditional validation procedure, affirming its ability to reproduce the radar’s perception in a validation dataset. Simple scenarios are employed to highlight the complete sensor model’s capabilities, illustrating the expected behavior for the specified scenarios.

The developed radar sensor model demonstrates effective modeling of radar cross-section (RCS) without detailed physical analysis, achieving satisfactory accuracy through statistical methods for replicating the RCS of moving vehicles. Its integration with an occlusion model exemplifies the potential for constructing object-list-based sensor models in a modular fashion. Such modularity is crucial for tailoring it to specific scenarios and Operational Design Domains (ODDs), where varying sensor effects are needed. The model’s flexibility allows for easy adaptation to new radar sensors, contingent upon the availability of suitable training data. The necessity of modular and adaptable sensor modeling approaches is underscored by the rapid evolution and diversity of sensors and their application scenarios.

In future research, this modeling technique will be extended to various datasets, utilizing diverse radar systems, particularly emphasizing the use of a 3-D radar that provides height information of cluster points. Furthermore, a more comprehensive and detailed validation of this radar model, along with the overall modeling methodology, will be conducted across a wider range of scenarios. This expanded approach aims to enhance the model’s applicability and accuracy in different scenarios and with varying technological setups.

## Figures and Tables

**Figure 1 sensors-24-02177-f001:**
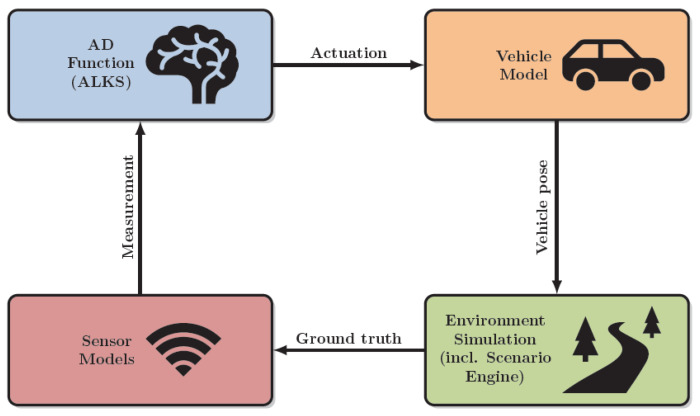
Simulation Framework for Virtual Validation of ADAS/AD functions, from [5].

**Figure 2 sensors-24-02177-f002:**
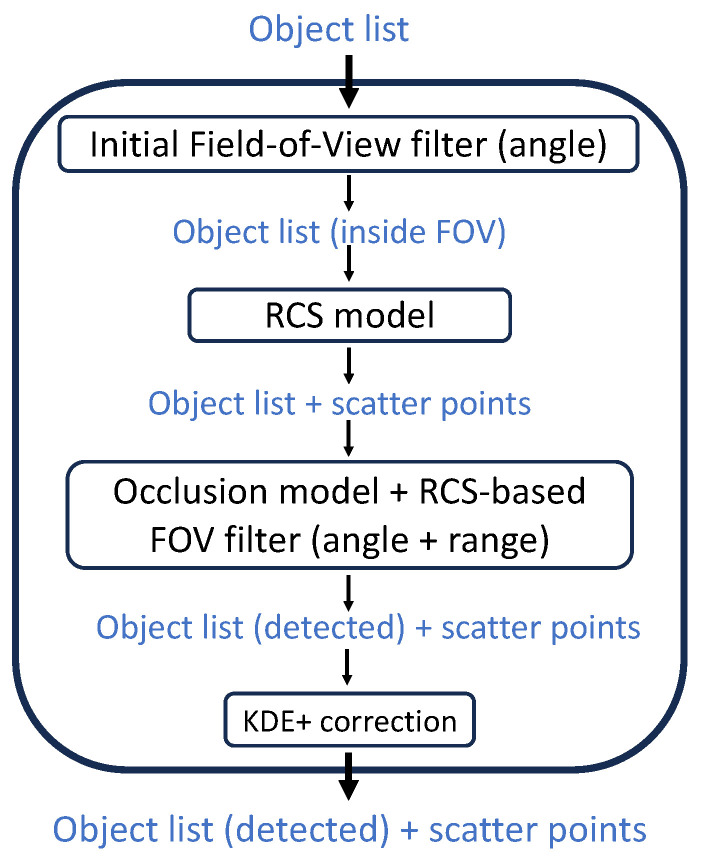
Components of the sensor model framework.

**Figure 3 sensors-24-02177-f003:**
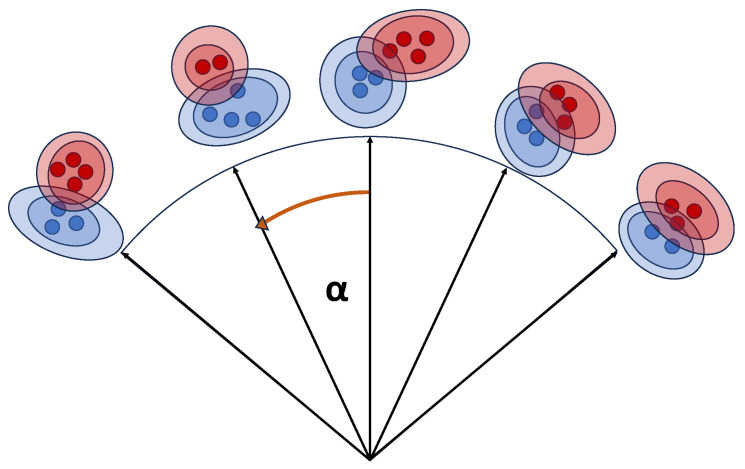
Schematic illustration of the model framework. The angular range is discretized by a fixed number of supporting points, each consisting of a Gaussian mixture model.

**Figure 4 sensors-24-02177-f004:**
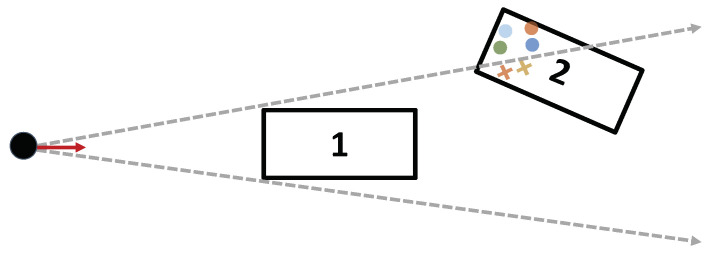
Example for an occlusion scenario. The object of interest (2) cuts in front of the occluder (1). The ego vehicle is indicated by the black circle, scatter points that are within the two arrows are considered to be undetected. The color represents the different RCS values.

**Figure 5 sensors-24-02177-f005:**
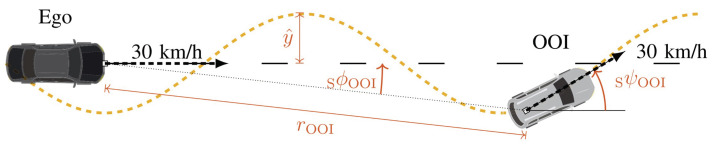
Schematic representation of the experimental setup. The azimuth angle and range are denoted as $s\phi_{OOI}$ and $r_{OOI}$. The difference in yaw angle is denoted as $\psi_{OOI}$ and the amplitude of the slalom as $\hat{y}$. Taken from [25].

**Figure 6 sensors-24-02177-f006:**
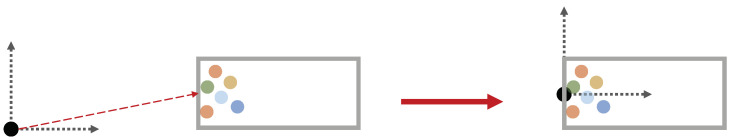
Coordinate transform into the OOI’s reference system. The center of the rear corresponds to the origin of the coordinate system. The colors represents the different RCS values.

**Figure 7 sensors-24-02177-f007:**
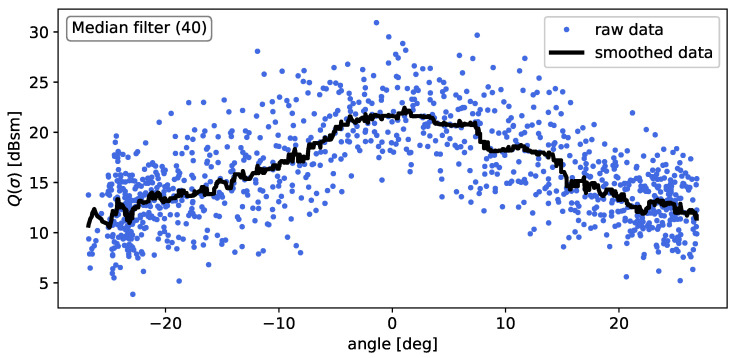
RCS profile of the vehicle. For better illustration, the data are smoothed using a moving median filter with a window size of 40.

**Figure 8 sensors-24-02177-f008:**
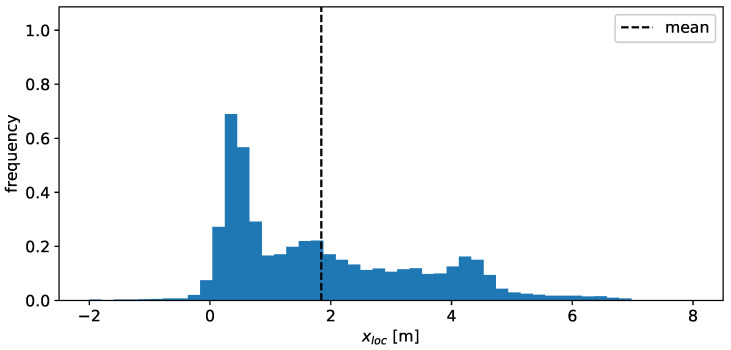
Histogram of the observed x-positions of the individual detections. A value of 0 corresponds to the rear of the vehicle.

**Figure 9 sensors-24-02177-f009:**
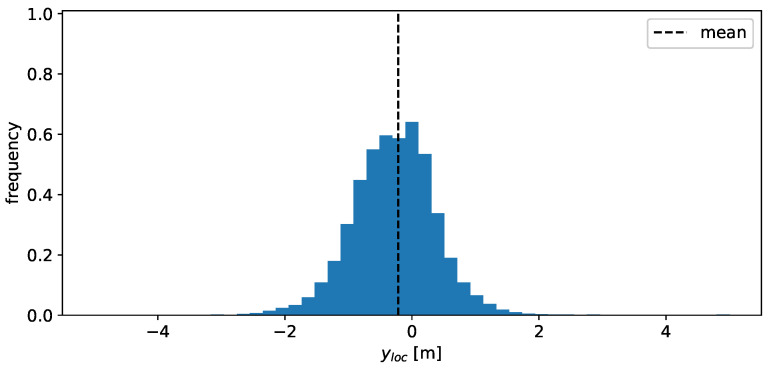
Histogram of the observed y-positions of the individual detections. A value of 0 corresponds to the center of the vehicle.

**Figure 10 sensors-24-02177-f010:**
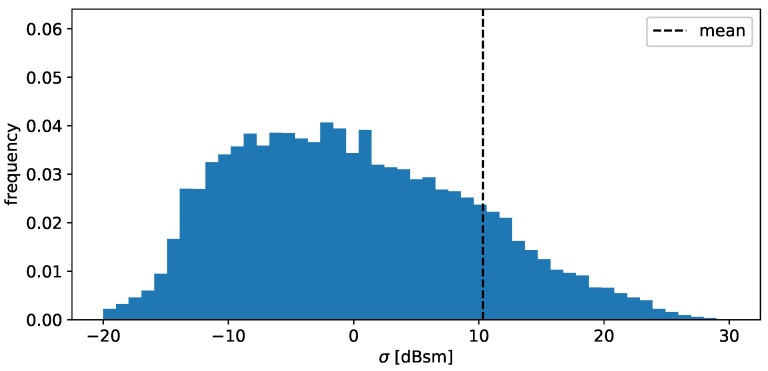
Histogram of the observed RCS values of the individual detections in decibel units. Note that due to the decibel units, the mean value appears to be shifted.

**Figure 11 sensors-24-02177-f011:**
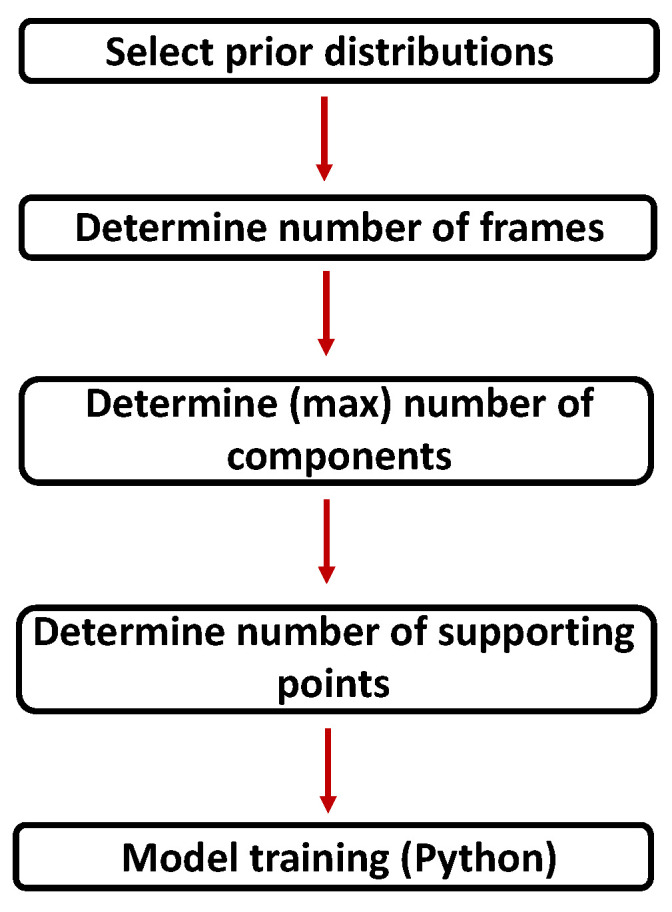
Flowchart illustrating the process of model creation. The individual steps are described in more detail in the following subsections.

**Figure 12 sensors-24-02177-f012:**
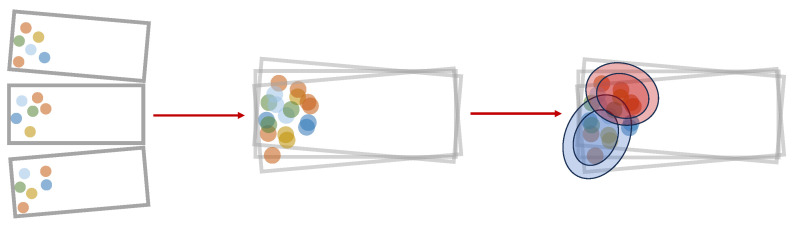
Augmentation of training data by grouping frames with similar aspect angles. The mean angle of the frames is assigned to the corresponding supporting point. The augmented training data are used for the training of Gaussian mixture models, depicted here by ellipses. The different colors of the points stands for the different RCS values.

**Figure 13 sensors-24-02177-f013:**
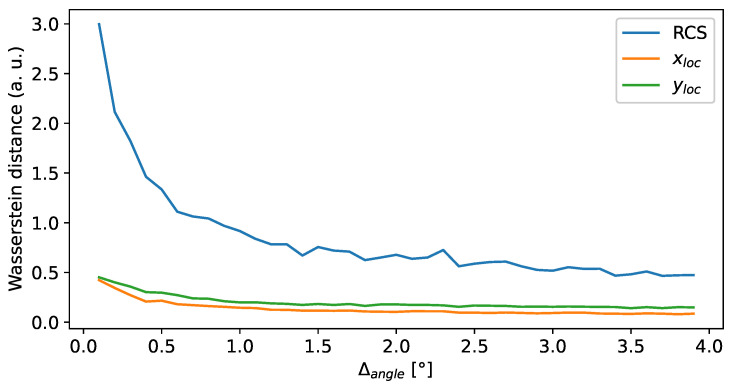
Wasserstein distance for the x, y, and RCS components over the angle interval.

**Figure 14 sensors-24-02177-f014:**
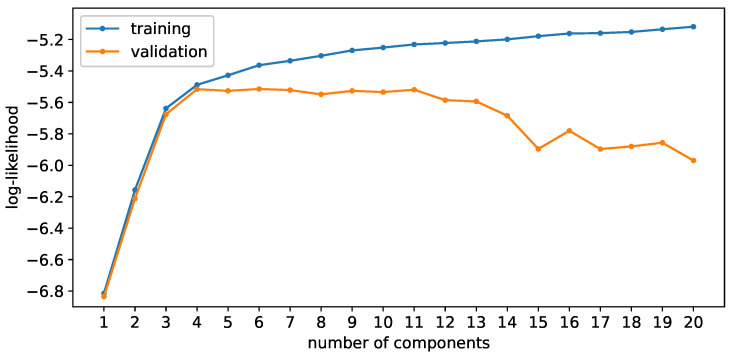
Log-likelihood during training (blue) and validation (orange) for the frequentist approach.

**Figure 15 sensors-24-02177-f015:**
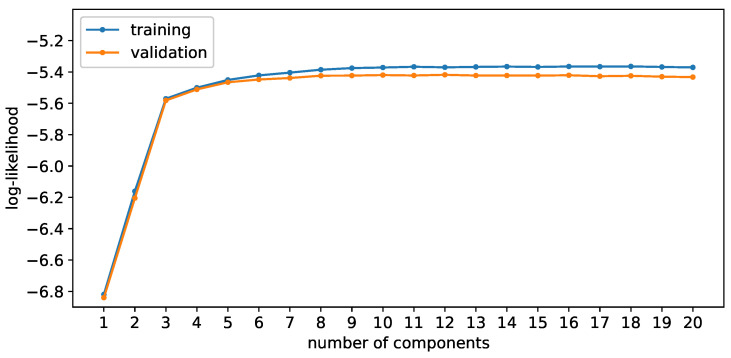
Log-likelihood during training (blue) and validation (orange) for the Bayesian approach.

**Figure 16 sensors-24-02177-f016:**
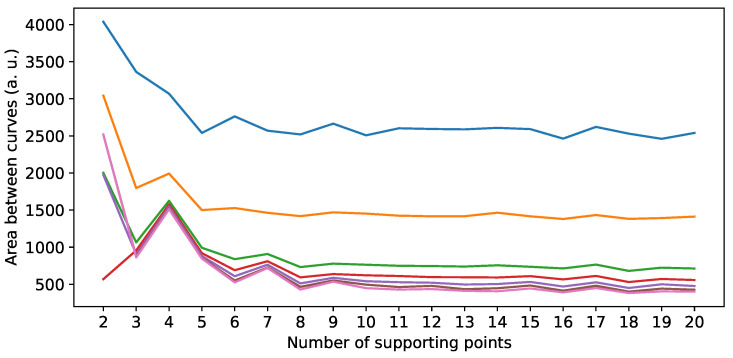
Area between curves over the number of supporting points. The different window sizes for the moving-median filter are depicted by the different lines. After about 8 supporting points, no significant improvement is achieved for additional supporting points, regardless of the window size.

**Figure 17 sensors-24-02177-f017:**
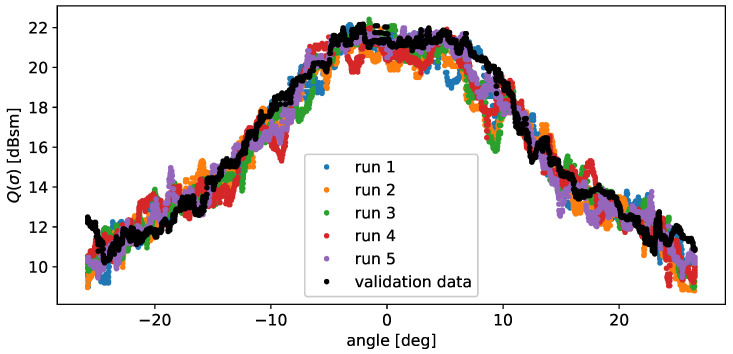
Comparison of the RCS characteristics of the validation data as well as the model output. A moving median filter with a window size of 40 was used for smoothing, and a total of five simulation runs were performed, indicated by different colours.

**Figure 18 sensors-24-02177-f018:**
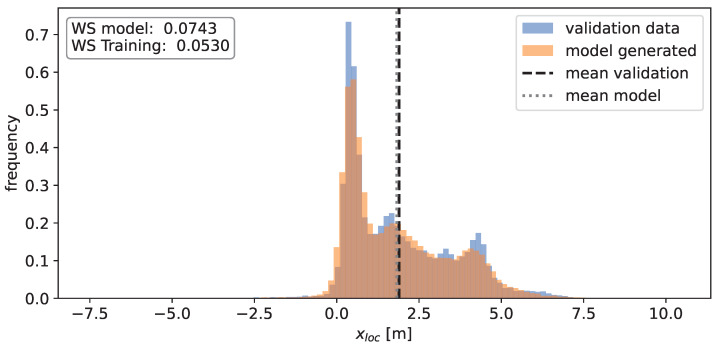
Histogram of the x-values for the model output (orange) and the validation data (blue). The means of the validation data (black) and the model-generated data (grey) are depicted as well. The values for the Wasserstein distance are depicted in the top left corner.

**Figure 19 sensors-24-02177-f019:**
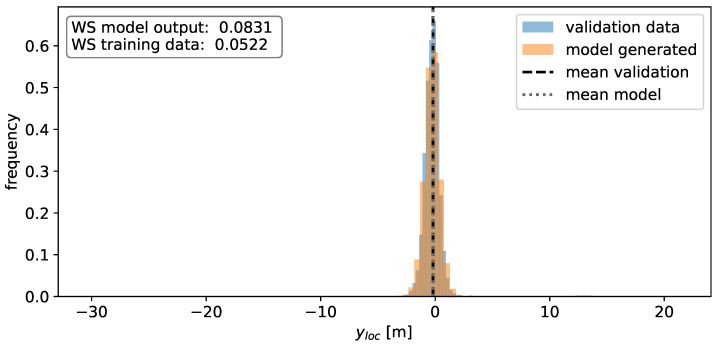
Histogram of the y-values for the model output (orange) and the validation data (blue). The means of the validation data (black) and the model-generated data (grey) are depicted as well. The values for the Wasserstein distance are depicted in the top left corner.

**Figure 20 sensors-24-02177-f020:**
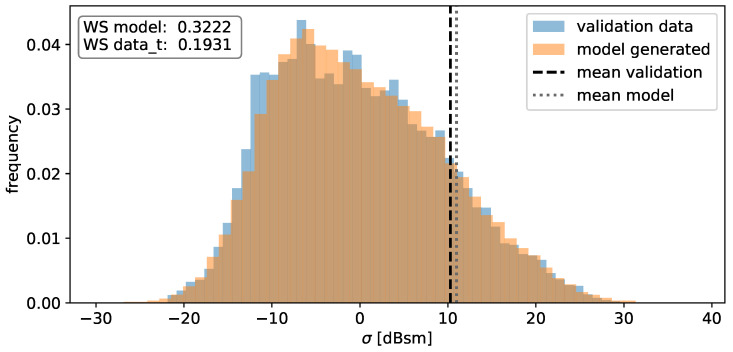
Histogram of the RCS-values for the model output (orange) and the validation data (blue). The means of the validation data (black) and the model-generated data (grey) are depicted as well. The values for the Wasserstein distance are depicted in the top left corner.

**Figure 21 sensors-24-02177-f021:**
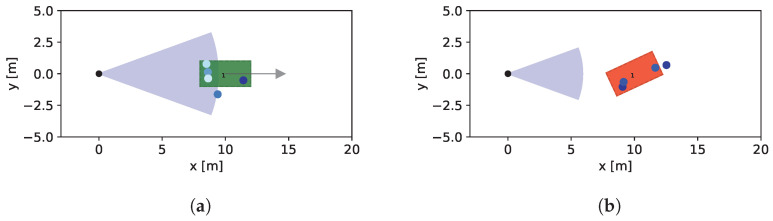
Field-of-View (FOV) for different relative orientations. Due to the strong angular dependence of the RCS, the effective FOV is strongly affected by the relative position of the vehicles. The green box on the left side represents an detected object and the red box on the right stands for an undetected object by the sensor. (**a**) Relative angle: 0°. (**b**) Relative angle: 25°.

**Figure 22 sensors-24-02177-f022:**
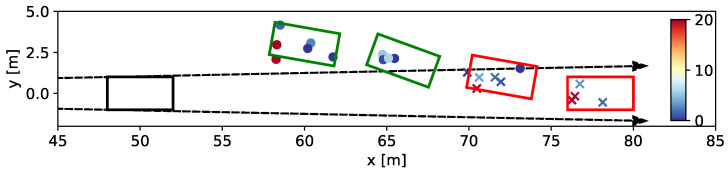
Example scenario, where a vehicle cuts in front of another vehicle. The graph shows four consecutive frames. All scatter points in between the arrows are occluded, indicated by crosses. The circles stands for visible cluster points. When the vehicle becomes sufficiently occluded, it is marked as undetected, shown with a red outline.

## Data Availability

No new data were created or analyzed in this study. Data are contained within the article.

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
