# Peer review of "Bayesian Gaussian Mixture Models for Enhanced Radar Sensor Modeling: A Data-Driven Approach towards Sensor Simulation for ADAS/AD Development"

_sensors, 2024, doi:10.3390/s24072177_

Round 1

Reviewer 1 Report

Comments and Suggestions for Authors

Aiming at modeling the radar cross section (RCS), the manuscript adopts a data-driven approach, using Gaussian Mixture Models to model the perception of radar. A Bayesian variational approach is used, and some simulations are performed to show the effectiveness of the proposed thought. However, I recommend making the following modifications.

1. Theoretical derivation is severely lacking. The theoretical feasibility of the proposed method has not been provided.

2. It says that "The model is expanded into a comprehensive radar sensor model based on object lists, incorporating occlusion effects and RCS-based detectability decisio", but where is the occlusion effect and detectability decisio expressed in the proposed method? How do they specifically execute?

3. Some figures do not have units for the vertical axis, such as figure 12 and figure 15.

4. The proposed method seems to only involve self-validation of effectiveness, lacking comparison methods and comparison with traditional existing methods, to show its advancement and innovativeness.

5. Regarding the specific applicable conditions, inapplicable conditions, advantages and disadvantages, required prior information, sensitivity to various environmental conditions, etc. of the proposed method, it is recommended to present them separately in a remark section.

Comments on the Quality of English Language

Minor editing.

Author Response

Many thanks for the constructive comments. We did an overall check and language editing of our paper to improve the readability.  Our answers to the individual comments can also be found in the order of the corresponding questions:

  1. The section covering the theoretical part was modified, now stating clearly that only the idea behind the derivation is given, and a reference is given where the reader can find the full derivation, including examples.)
  2. The occlusion model and RCS-based detectability are described in section 2.3.
  3. Figures were adapted, stating ‘(a.u.)’ since there the units are arbitrary. For the plots with the log-likelihood and histograms, no unit was stated since it does not have any units.
  4. For comparison, more data, which is currently not available, would be needed. This issue is now clearly stated in the final section.
  5. As suggested, a remark section was added at the end, describing the current limitations.

Reviewer 2 Report

Comments and Suggestions for Authors

This paper proposes a data-driven approach, using Gaussian Mixture Models (GMMs) to model the radar’s perception for various vehicles and aspect angles. A Bayesian variational approach automatically infers model complexity. The model is expanded into a comprehensive radar sensor model based on object lists, incorporating occlusion effects and RCS-based detectability decisions. The model’s effectiveness is demonstrated through accurate reproduction of the RCS behavior and scatter point distribution. The comments are listed below.

1.        The necessity and reason of proposed approach need to be further enhanced in the introduction.

2.        The data and training part is introduced in detail, suggest adding a flowchart of all training process to improve clarity to reader.

3.        Table 1 title has to many information, suggest put in the paragraph nearby and replaced with a simple title.

4.        The training part seem use a RCS list as reference, the data and acquisition of this data should be introduced.

Comments on the Quality of English Language

correct minor grammar and spell error 

Author Response

Many thanks for the constructive comments. We did an overall check and language editing of our paper to improve the readability.  Our answers to the individual comments can also be found in the order of the corresponding questions:

  1. The introduction was carefully revised and a more detailed description of the reasons and contributions of the study was added in section 1.2.
  2. A flowchart was added for a better overview.
  3. Information is now in the next and the table is removed.
  4. The training and the validation is only based on the experimental data that is described in section 3.

Reviewer 3 Report

Comments and Suggestions for Authors

After carefully reading this article, especially the core content, I find it difficult to discover its advantages, which are reflected in the following aspects: 1. In the radar sensor model section, the author only summarizes the basic radar principles, including filtering, RCS, and clustering, which are conventional and widely existing basic knowledge for many years, and cannot be a main contribution. 2. The author claims to have proposed the Bayesian GMM method, but from the content of 2.2.2, it can be seen that Bayesian method and GMM are completely separated, and the author has only used the most classic Bayesian method without making any substantial improvements or innovations. The calculation of the RCS module is still a variant of the radar equation, and its parameterization has not been modeled. The significance of using media filtering in Figure 7 is not significant, as radar data is affected by a lot of noise and clutter, and the basic filtering operation performance is poor. The use of the mean method to illustrate Figures 8-9 is also the most basic, without in-depth analysis of their data.

Author Response

Many thanks for the constructive comments. We did an overall check and language editing of our paper to improve the readability.  Our answers to the individual comments can also be found in the order of the corresponding questions:

  • In the radar sensor model section, the author only summarizes the basic radar principles, including filtering, RCS, and clustering, which are conventional and widely existing basic knowledge for many years, and cannot be a main contribution.

Answer: The main contribution is meant to be the use of Bayesian Gaussian Mixture Models to model the spatial distribution of RCS in order to enable a more accurate modelling of occlusion effects without costly simulations. Most known Radar models were either modeling the RCS on an object level (i.e. the total RCS is assigned to one single point) or using computationally expensive (and therefore slow) calculations.

  • The author claims to have proposed the Bayesian GMM method, but from the content of 2.2.2, it can be seen that Bayesian method and GMM are completely separated, and the author has only used the most classic Bayesian method without making any substantial improvements or innovations.

Answer: It is indeed true that this particular use of Bayesian Gaussian Mixture Models is not new. However, the use of it for the modelling of the radar’s perception is new. Using the Bayesian approach avoided overfitting which was a common and hard-to-solve problem with the frequentist approach.

  • The calculation of the RCS module is still a variant of the radar equation, and its parameterization has not been modeled. 

Answer: The use of the radar equation is only meant to be an illustrative example of how to make use of the RCS modelling approach. Due to its modularity, the radar equation could simply be replaced by a more sophisticated approach, for example semiempirical methods. This is now stated in the remarks section as well.

  • The significance of using media filtering in Figure 7 is not significant, as radar data is affected by a lot of noise and clutter, and the basic filtering operation performance is poor. 

Answer: The filter is only applied for illustrative purposes and does not have any quantitative significance, as you correctly pointed out.

  • The use of the mean method to illustrate Figures 8-9 is also the most basic, without in-depth analysis of their data.

Answer: The mean is only an additional information, the use of the Wasserstein distance provides more details about the similarity of the distributions. 

Round 2

Reviewer 1 Report

Comments and Suggestions for Authors

In general, the Remark section is placed before the Conclusion section. Then, the manuscript can be considered to be accepted.

Comments on the Quality of English Language

The English language should be carefully checked and made minor revisions.

Author Response

We have made the necessary update for moving the remark before the conclusion as recommended. Thank you for your positive feedback. 

Kind regards!

Reviewer 3 Report

Comments and Suggestions for Authors ALL my concerns have been fixed.

Author Response

We thank the reviewer for the positive feedback!